# Cobalt catalyzed practical hydroboration of terminal alkynes with time-dependent stereoselectivity

Jinglan Wen[1,2], Yahao Huang[1,2], Yu Zhang[1,2], Hansjörg Grützmacher [2,3] & Peng Hu [1,2,4] ✉

Stereodefined vinylboron compounds are important organic synthons. The synthesis of $E$−1-vinylboron compounds typically involves the addition of a B-H bond to terminal alkynes. The selective generation of the thermodynamically unfavorable $Z$-isomers remains challenging, necessitating improved methods. Here, such a proficient and cost-effective catalytic system is introduced, comprising a cobalt salt and a readily accessible air-stable CNC pincer ligand. This system enables the transformation of terminal alkynes, even in the presence of bulky substituents, with excellent $Z$-selectivity. High turnover numbers (>1,600) and turnover frequencies (>132,000 h$^{-1}$) are achieved at room temperature, and the reaction can be scaled up to 30 mmol smoothly. Kinetic studies reveal a formal second-order dependence on cobalt concentration. Mechanistic investigations indicate that the alkynes exhibit a higher affinity for the catalyst than the alkene products, resulting in exceptional $Z$-selective performance. Furthermore, a rare time-dependent stereoselectivity is observed, allowing for quantitative conversion of $Z$-vinylboronate esters to the $E$-isomers.

Organoboron compounds are remarkable versatile building blocks for organic synthesis due to their availability and exceptional tolerance toward various functional groups[1–6]. Among these compounds, alkenylboron compounds with a defined stereochemistry play a crucial role in a wide range of cross-coupling reactions. These reactions lead to the generation of functionalized alkenes possessing specific configurations, which are important for the synthesis of bioactive compounds and natural products[7–20]. The hydroboration of terminal alkynes represents a straightforward and economically efficient method to generate vinylboron compounds. The anti-Markovnikov addition of the B-H group to the C≡C bond of terminal alkynes primarily affords $E$-1-alkenylboron compounds as the major products[8–32]. Conversely, achieving the production of $Z$-alkenylborones from terminal alkynes via hydroboration presents a more challenging task, primarily due to

the fact that the $E$-isomers are thermodynamically more stable. Consequently, only a dozen of catalytic reactions leading selectively to the $Z$-isomers through hydroboration of terminal alkynes have been reported[31–44]. Notably, these studies predominantly employ noble metal catalysts, including rhodium[34,35,44], iridium[34], ruthenium[36,42], and palladium[37,40]. Among these catalytic systems, the ruthenium pincer catalyst reported by the Leitner group stands out as the most versatile and active one, displaying moderate turnover numbers (TON) of up to 920 and turnover frequencies (TOF) of up to 38 h$^{-1}$ at −15 °C[36]. To date, the highest TON of 9800 was achieved by Saito et al. using a ruthenium complex with an $N$-heterocyclic carbene and PCy$_3$ ligand but this reaction necessitated a long reaction time of 6 days and heating[42].

In the pursuit of developing more sustainable catalysts, the exploration of complexes with earth-abundant metals has been in

[1]Institute of Green Chemistry and Molecular Engineering, School of Chemistry, Sun Yat-sen University, Guangzhou 510006, PR China. [2]Lehn Institute of Functional Materials, School of Chemistry, Sun Yat-sen University, Guangzhou 510006, PR China. [3]Department of Chemistry and Applied Biosciences, ETH Zürich, 8093 Zürich, Switzerland. [4]State Key Laboratory of Structural Chemistry, Fujian Institute of Research on the Matter, Chinese Academy of Sciences, Fuzhou 350002 Fujian, PR China. ✉e-mail: hupeng8@mail.sysu.edu.cn

the focus of recent research[45–54], which includes the investigation of Z-selective hydroboration of terminal alkynes[38,39,41,43]. To date, three pincer Co and Fe complexes have been developed to facilitate this transformation with favorable Z-selectivity (Fig. 1a). Pioneering work by Chirik et al. demonstrated the possibility to perform the Z-selective hydroboration of alkyl and aryl alkynes at room temperature using a bis(imino)pyridine cobalt catalyst[38]. Kirchner et al. developed an iron polyhydride catalyst [Fe(PNP)(H)$_2$(η$^2$-H$_2$)] with a non-classical hydrogen ligand for the hydroboration reaction in C$_6$D$_6$, demonstrating moderate TONs of up to 245 and TOFs of up to 82 h$^{-1}$[41]. Furthermore, de Ruiter et al. recently reported the use of a related iron complex [Fe(PC$_{NHC}$P)(H)$_2$(N$_2$)] with a labile dinitrogen ligand as catalyst precursor for performing this type of reaction[43].

Despite the existence of successful catalytic examples demonstrating highly Z-selective hydroboration of terminal alkynes, the catalytic systems reported so far often suffer from limitations in terms of substrate scope, stereoselectivity, and reactivity toward sterically hindered alkyl and/or aryl alkyne substrates. For instance, arylacetylenes with substituents at ortho or even meta positions on aryls proved to be challenging substrates. Achieving good Z-selectivity typically requires the use of ligands and complexes synthesized in multiple-steps, which hinders their practical application in syntheses[31–44]. Typically, bulky pincer ligands with flexible arms are employed to stabilize the complexes and improve catalytic performance. However, the use of such bulky complexes inevitably leads to low efficiencies in catalytic reactions with sterically encumbered substrates. Given the limitations of the current methods on one side but the broad use of vinylboronate esters on the other, there is a high need for the development of a general, cost-effective, highly-efficient, and practical synthetic route for the stereo-selective hydroboration of terminal alkynes.

## Results

In this work, we present a catalytic system utilizing a cobalt salt and a CNC-$^i$Pr ligand as precursors to generate stereoselective hydroboration catalysts (Fig. 1b). The CNC-$^i$Pr ligand was synthesized following a procedure previously reported by Herrmann with modifications[55]. By employing commercially inexpensive compounds, we successfully synthesized the air-stable salt CNC-$^i$Pr in one-step with an isolated yield of 99% (Fig. 1c). The flat structure of the CNC-pincer ligand—obtained from the salt in situ by double deprotonation—with two isopropyl groups in the periphery pointing away from the metal center should allow to coordinate a wide range of terminal alkynes, including those with sterically demanding substituents that were previously challenging to functionalize selectively. Indeed, this catalytic system exhibits an exceptional TON of up to 1680, surpassing the performance of noble metal catalysts at room temperature. Additionally, TOFs of up to 132,480 h$^{-1}$ were reached. To the best of our knowledge, this represents the fastest transition metal-catalyzed hydroboration reaction reported to date. Also, a rare time-dependent stereoselectivity was observed which allows to prepare either the Z-isomer or E-isomer only by variation of the reaction time.

The initial investigation focused on utilizing different NHC-pincer ligands in conjunction with CoCl$_2$ to achieve Z-selective hydroboration of phenylacetylene (1a) in DMF at room temperature (Table 1). A significant influence of the structure of the ligands on the reaction outcome was observed (entries 1–6). Notably, the use of the CNC-$^i$Pr ligand resulted in the formation of styrylboronate ester (1b) with a remarkable 99% yield and high E/Z selectivity of 98:2 (entry 4). However, when ligands with smaller (entries 1–3) or bulkier N-substituents (entries 5, 6) were employed, the reactions gave lower or no yields and also lower E/Z selectivities. To our surprise, we discovered that reducing the reaction time to just 10 min led to the formation of Z-styrylboronate ester in 99% yield with an inverse Z/E selectivity of 96:4 (entry 7). This result suggests the possibility of a Z/E isomerization process during the reaction and indicates that the stereoselectivity of

hydroboration may be kinetically controlled (entry 7 vs. 4). Interestingly, reducing the amount of pinacol borane, HBpin, to 1.5 equivalents significantly suppressed the isomerization process (entry 8 vs. 4). Furthermore, substituting CoCl$_2$ (entry 9) with Co(acac)$_2$ proved to give a more efficient catalyst in combination with CNC-$^i$Pr as ligand (entry 10). After further optimization (Supplementary Tables 9–12), the reaction could be successfully carried out with excellent Z-selectivity, achieving a 90% isolated yield at a low catalyst loading of 0.1 mol% (entry 11). Notably, reducing the amount of HBpin did not affect the performance of the reaction (entry 12).

Subsequently, we investigated the applicability of our catalytic system to various terminal alkynes for Z-selective hydroboration (Fig. 2). Delightfully, very high selectivity and moderate to excellent yields for all the tested substrates were observed (1a–42a). Arylacetylenes bearing electron-donating groups on the para-position of the benzene ring (1a–6a vs. 8a–12a) generally exhibited higher yields of Z-styrylboronate esters (1b–6b) compared to those with electron-withdrawing groups (8b–12b). Aryl alkynes (13a–16a) with meta-substituted phenyl groups displayed no diminished reactivity or selectivity. Ortho-substituted Z-styrylboronate esters (17b–22b) could be obtained in high yields with excellent Z-selectivity from the corresponding arylacetylenes, which are typically challenging substrates due to steric hindrance. Surprisingly, even sterically highly demanding phenylacetylenes with two ortho-substituents on the phenyl ring, such as 2,6-diisopropylphenylacetylene (26a), exhibit excellent Z-selectivity and yields of Z-styrylboronate esters (23b–26b) are high (>80%). These results highlight the excellent compatibility of the Co(acac)$_2$/CNC-$^i$Pr catalytic system with sterically hindered substrates. Further exploration of arylacetylene substrates revealed good tolerance toward thienyl (27b, 28b) and polycyclic aryl (29b) groups. Interestingly, even a substrate bearing two ethynyl groups smoothly underwent the reaction, leading to the target product with an impressive 82% isolated yield and very high ZZ selectivity (30b). Importantly, we also examined the possibility of conducting reactions with a low catalyst loading of

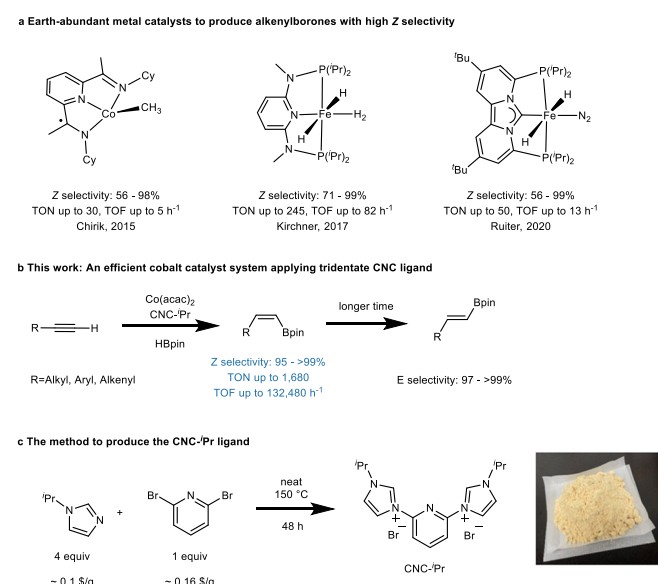

**a** Earth-abundant metal catalysts to produce alkenylborones with high Z selectivity

Z selectivity: 56 - 98%
TON up to 30, TOF up to 5 h$^{-1}$
Chirik, 2015

Z selectivity: 71 - 99%
TON up to 245, TOF up to 82 h$^{-1}$
Kirchner, 2017

Z selectivity: 56 - 99%
TON up to 50, TOF up to 13 h$^{-1}$
Ruiter, 2020

**b** This work: An efficient cobalt catalyst system applying tridentate CNC ligand

R=Alkyl, Aryl, Alkenyl

Z selectivity: 95 - >99%
TON up to 1,680
TOF up to 132,480 h$^{-1}$

E selectivity: 97 - >99%

**c** The method to produce the CNC-$^i$Pr ligand

4 equiv
~ 0.1 $/g

1 equiv
~ 0.16 $/g

neat
150 °C
48 h

CNC-$^i$Pr
Isolated yield: 99%, 14.3 g

**Fig. 1 | Selective hydroboration of terminal alkynes catalyzed by earth-abundant metals. a** Comparison of three reported earth-abundant metal catalysts for achieving highly Z-selective hydroboration of terminal alkynes. The Z selectivity, highest turnover numbers (TONs), and turnover frequencies (TOFs) of each system are shown. **b** This work: an efficient and general cobalt catalyst system for hydroboration of terminal alkynes with time-dependent stereoselectivity. **c** Easy large-scale production of CNC-$^i$Pr from inexpensive commercially available compounds with high isolated yields. Prices listed are sourced from bidepharm.com.

**Table 1 | Optimization study**

Reaction scheme: 1a + H-Bpin → (Co precursor, Ligand, tBuOK, DMF, N₂, r.t., t) → 1b ($B$ = Bpin)

Ligands: CNC-H; CNC-R where R = Me, CNC-Me; R = Et, CNC-Et; R = $^i$Pr, CNC-$^i$Pr; R = $^t$Bu, CNC-$^t$Bu; R = mesityl, CNC-Mes

| Entry | Precursor | Ligand | Yield (%)[a] | Z/E ratio[b] |
|---|---|---|---|---|
| 1 | CoCl$_2$ | CNC-H | 0 | – |
| 2 | CoCl$_2$ | CNC-Me | 71% | 3:97 |
| 3 | CoCl$_2$ | CNC-Et | 78% | 36:64 |
| 4 | CoCl$_2$ | CNC-$^i$Pr | 99% | 2:98 |
| 5 | CoCl$_2$ | CNC-$^t$Bu | 7% | <1:99 |
| 6 | CoCl$_2$ | CNC-Mes | 0 | – |
| 7[c] | CoCl$_2$ | CNC-$^i$Pr | 99% | 96:4 |
| 8[d] | CoCl$_2$ | CNC-$^i$Pr | 70% | >99:1 |
| 9[e] | CoCl$_2$ | CNC-$^i$Pr | 45% | 97:3 |
| 10[e] | Co(acac)$_2$ | CNC-$^i$Pr | 98% | 51:49 |
| 11[f] | Co(acac)$_2$ | CNC-$^i$Pr | 99% (90%)[g] | 98:2 |
| 12[f,h] | Co(acac)$_2$ | CNC-$^i$Pr | 98% (90%)[g] | >99:1 |

Reaction conditions: CoCl$_2$ (5.0 mol%), **1a** (0.4 mmol, 1.0 equiv), ligand (5.0 mol%), $^t$BuOK (20.0 mol%), HBpin (3.0 equiv), DMF (2 ml), room temperature (r.t.), N$_2$ atmosphere, 12 h.
[a]NMR yield determined using methylene bromide as the internal standard.
[b]Z/E ratio based on $^1$H NMR analysis.
[c]Reacted for 10 min.
[d]HBpin used in 1.5 equiv., 24 h.
[e]Co precursor (0.2 mol%), ligand (0.22 mol%), $^t$BuOK(0.8 mol%), DMF (1 ml).
[f]Co(acac)$_2$ (0.1 mol%), ligand (0.14 mol%), $^t$BuOK(0.56 mol%), DMF (0.5 ml).
[g]Isolated yield.
[h]HBpin (1.3 equiv.).

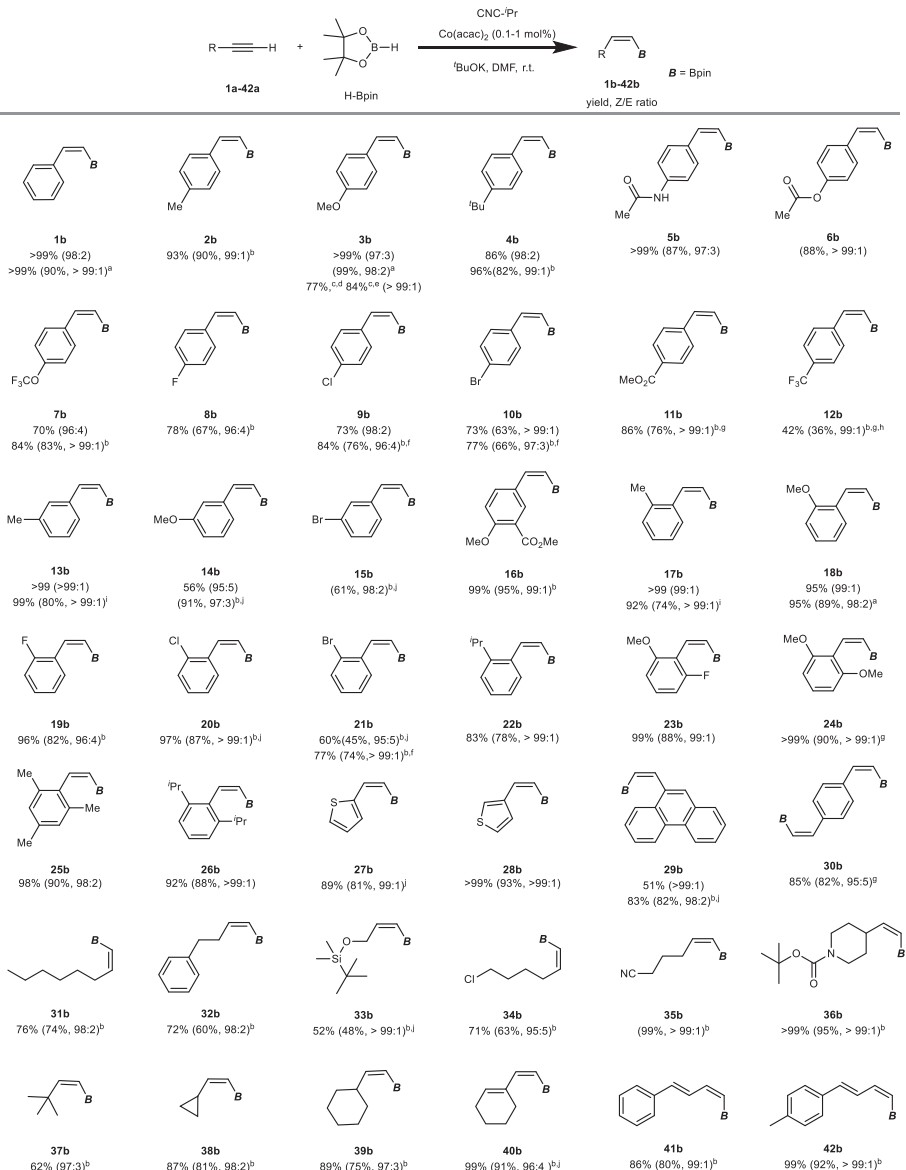

**Fig. 2 | Scope of *Z*-selective hydroboration of terminal alkynes.** Reaction conditions: terminal alkyne (1.0 equiv, 0.4 mmol), HBpin (1.3 equiv), Co(acac)$_2$ ([Co], 0.5 mol%), CNC-*i*Pr (1.4 equiv to [Co]), *t*BuOK (5.6 equiv to [Co]) in DMF (0.5 ml) at room temperature (r.t.). See Supplementary Information for experimental details. $^1$H NMR yields are shown with methylene bromide or mesitylene as the internal standard. Isolated yields and *Z/E* ratios are provided in parenthesis. [a]0.1 mmol% Co(acac)$_2$, 0.8 mmol scale. [b]1 mmol% Co(acac)$_2$. [c]0.05 mmol% Co(acac)$_2$, yield determined by GC-MS. [d]24 h. [e]48 h. [f]2.5 equiv. HBpin. [g]3 equiv. HBpin. [h]With 10 mol% diphenylacetylene. [i]0.2 mmol% Co(acac)$_2$. [j]2 equiv. HBpin.

0.05 mol% and with *p*-methoxystyrene **3a** we achieved yields of 77% (TON = 1540) and 84% (TON = 1680) in 24 and 48 h, respectively (**3b**).

The compatibility of the Co(acac)$_2$/CNC-*i*Pr catalytic system with alkyl (**31a**–**39a**) and alkenyl alkynes (**40a**–**42a**) was investigated. Similar to the sterically hindered arylacetylene substrates, *tert*-butyl and cyclic alkyl/alkenyl functionalized terminal alkynes were also efficiently converted into the desired vinylboronate esters (**36b**–**40b**) with excellent *Z*-selectivity. This unique capability of the catalytic system to hydroborate both aryl and alkyl substituted terminal alkynes, particularly those containing bulky substituents with overall very high *Z*-selectivity (*Z:E* > 95:5), is not found in the literature. Importantly, the reaction exhibits a broad substrate scope, accommodating various important functional groups, thereby providing flexibility for subsequent transformations.

To assess the practicality of our procedure, we conducted large-scale reactions under an inert atmosphere (nitrogen). Initially, an ice bath was employed to regulate the temperature due to the exothermic nature of the reaction. Figure 3 illustrates the results obtained from reactions on the 30 mmol scale, which proceeded smoothly and rapidly, yielding various *Z*-alkenylboronate esters **1b**, **3b**, **13b**, **22b**, **28b**, **31b**, and **38b** with good to excellent isolated yields and selectivity. Most reactions at the 30 mmol scale gave even better results when compared to the small-scale reactions. Significantly, the utilization of compound **3b** has been demonstrated in the synthesis of the natural product Nyasol[19]. This methodology holds promise for the efficient large-scale production of valuable *Z*-alkenylboronate esters[8–19,31,32].

## Discussion

The performance of the Co(acac)$_2$/CNC-*i*Pr catalytic system prompted us to revisit the optimization study (vide supra) with the aim to better understand under which conditions *Z* or *E* selectivity can be achieved (Table 1, entries 4, 7). Typically, obtaining good *Z/E* stereoselectivity in the synthesis of alkenylboron compounds requires the use of two different catalyst systems, each favoring either a kinetic or

thermodynamically controlled process. However, with the Co(acac)₂/CNC-*i*Pr catalyzed reactions it is possible to produce both isomers selectively using the same catalyst system at room temperature but at different reaction times. This finding indicates a significantly faster rate for the formation of the *Z* isomer compared to the *E* isomer (Fig. 4a, $k_1 \gg k_2$, $k_3$). Additionally, we observed a relatively slow but nearly irreversible *Z/E* isomerization process ($k_3 \gg k_{-3}$), leading predominantly to

the *E*-configured product after a longer reaction time. To gain further insights into the dynamic progress of the reaction, we monitored the amounts of **1a**, **Z-1b**, and **E-1b** over time (Fig. 4b). Using 0.5 mol% catalyst and 2 equivalents of HBpin, the reaction exhibited an induction period of ~70 s, indicating the formation of the catalytically active species. Subsequently, there was a rapid consumption of **1a** within 90 s, leading to an almost quantitative formation of **Z-1b** and only negligible amounts of **E-1b** were observed. Over the next 5 h, **Z-1b** gradually transformed into **E-1b** at a nearly constant rate (see Supplementary Fig. 1). This phenomenon suggests that the increasing concentration of **E-1b** does not affect the transformation rate, which differs from traditional reversible isomerization processes. Importantly, it is this feature of the Co(acac)₂/CNC-*i*Pr catalytic system that explains the formation of **E-1b** with an excellent *E*-stereoselectivity (large *E/Z* ratio).

Further analysis revealed that the reaction exhibited a pronounced kinetic response to both the catalyst concentration and the amount of HBpin. Therefore, the increased presence of HBpin can significantly promote the reaction process due to kinetic effects (Supplementary Table 4, see also Fig. 2). Variation of the catalyst concentration has a more significant influence. To investigate the kinetics of the reaction, we examined the apparent reaction rate constants ($k_{obs}$) at different catalyst concentrations ([Co]) (Fig. 4c and Supplementary Fig. 2). Remarkably, plotting log($k_{obs}$) against log[Co] yielded a straight line with a slope of 2.30, indicating an approximately second-order dependence of the reaction on the catalyst

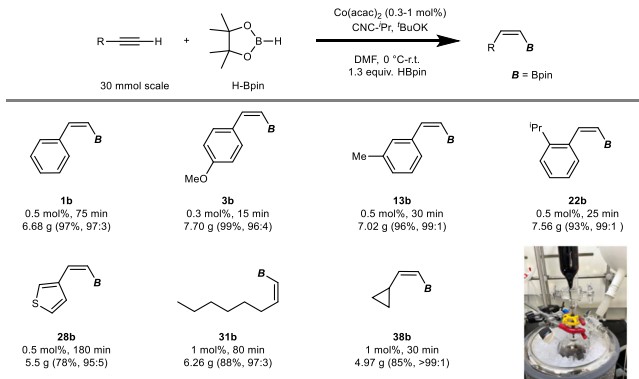

**Fig. 3 | Large scale reaction.** Reaction conditions: terminal alkyne (1.0 equiv, 30 mmol), HBpin (1.3 equiv), Co(acac)₂ ([Co], as presented), CNC-*i*Pr (1.4 equiv to [Co]), *t*BuOK (5.6 equiv to [Co]) in DMF (35 ml) at 0 °C–room temperature (r.t.).

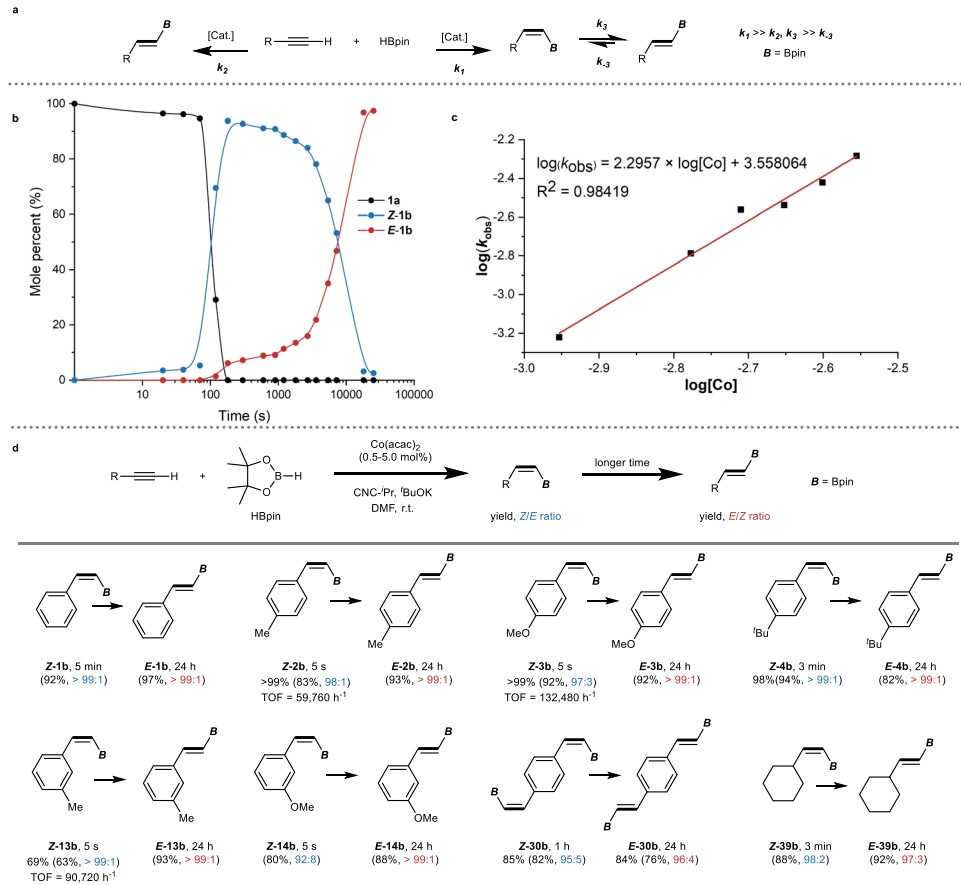

**Fig. 4 | Kinetic investigation. a** Proposed equilibrium equation of the hydroboration of terminal alkynes. **b** Kinetic profile of the hydroboration of phenylacetylene (**1a**). Reaction conditions: **1a** (1.0 equiv, 0.4 mmol), HBpin (2.0 equiv), Co(acac)₂ ([Co], 0.5 mol%), CNC-*i*Pr (1.4 equiv to [Co]), *t*BuOK (5.6 equiv to [Co]) in DMF (0.5 ml) at room temperature (r.t.). **c** Kinetic analysis of the formal reaction order based on the concentration of catalyst ([Co]). **d** Time-dependent stereoselective hydroboration of

terminal alkynes. Reaction conditions: terminal alkyne (1.0 equiv, 0.4 mmol), HBpin (3.0 equiv), Co(acac)₂ ([Co], 0.5–5 mol%), CNC-*i*Pr (1.4 equiv to [Co]), *t*BuOK (5.6 equiv to [Co]) in DMF (0.5 ml) at room temperature (r.t.). See Supplementary Information for experimental details. ¹H NMR yields are shown with methylene bromide as the internal standard. Isolated yields and *Z/E* ratios in parenthesis. The reaction time for the transformation of the *E* isomers was not optimized.

concentration ([Co]). This result is in agreement with the experiments that showed larger reaction rates at higher catalyst concentrations.

Based on the kinetic studies, a time-dependent method for the stereoselective hydroboration of terminal alkynes was successfully developed (Fig. 4d). Apart from substrate **1a**, numerous alkyne substrates were effectively converted into their corresponding *Z*- and *E*-alkenylboronate esters selectively within varying reaction times, utilizing the same reaction setup. Notably, both isomers exhibited very high stereoselectivity. Note that the catalyst derived from Co(acac)$_2$/CNC-$^i$Pr shows the fastest rates ever reported for *Z*-selective hydroboration reactions, with certain reactions completing instantaneously upon mixing the reactants and catalyst, leading to impressive TOFs for the production of some *Z*-alkenylboronate esters (see **Z-2b**, **Z-3b**, **Z-13b**).

Deuterium labeling experiments were performed in order to gain a deeper understanding of a possible reaction mechanism. The hydroboration of 1-(4-methoxyphenyl) acetylene-2-*d* (**3a**-*d*) produced **Z-3b**-*d* with 99% deuterium retention at the *beta* position of the Bpin group (Fig. 5a, eq 1 left and Supplementary Fig. 3). This result is consistent with the 1,1-hydroboration process observed in the catalytic hydroboration of alkynes by noble metals (Rh, Ir, Ru)[34–36] and earth-abundant metals (Co, Fe)[38,41] as previously reported. With 1 mol% Co(acac)$_2$/CNC-$^i$Pr and 1 equiv of HBpin, isomerization of **Z-3b**-*d* to **E-3b**-*d* occurred, resulting in a reduced deuteration percentage (from >99% to 46%) at the *beta*-position of the Bpin group (Fig. 5a, eq 1 right and Supplementary Fig. 4). Prolonging the reaction time did not alter this deuteration ratio, and no isotopic labeling was observed geminal to the BPin group. These findings suggest that a Co-H insertion followed by a *β*-H elimination occurs during the isomerization process[56]. An appropriate excess of HBpin can ensure the generation of a sufficient concentration of the Co-H complex and provide a reductive environment to stabilize it. Similar reactions applying **3a** and DBpin were also performed, leading to the same conclusion as Fig. 5a, eq 1 (Supplementary Figs. 5–7). This observation is distinct from the reactions performed by Chirik et al.[38]. When a 1:1 mixture of 1-phenylacetylene-2-*d* (**1a**-*d*) and 4-methoxy-phenylacetylene (**3a**) was employed, both corresponding hydroboration products exhibited ~50% deuterium exchange, indicating intermolecular hydrogen transfer of the C(sp)-H bonds (Fig. 5a, eq 2 and see Supplementary Figs. 8 and 9). We also conducted kinetic isotope effect experiments on the C(sp)-H bond and B-H bond, and observed similar KIE values (1.07 and 0.96, respectively, see Supplementary Figs. 10 and 11), suggesting that activation of either bond is not the rate-determining step (Fig. 5a, eq 3 and 4). Furthermore, the different deuterium-labeled *Z*-alkenylboronate esters clearly demonstrate that the hydrogen atom adjacent to the Bpin group originates from HBpin, while the hydrogen at the *beta* position is transferred from the alkynyl hydrogen, in accordance with eq 1 (Fig. 5a, eq 3 and 4).

Subsequently, a mixture of **3a** and 0.1 equiv of 1,2-diphenylethyne, which remained inert at room temperature, was prepared using a 0.5 mol% catalyst system with 3 equiv of HBpin to investigate the influence of an alkyne on the *Z/E* isomerization process (Fig. 5b). In the absence of 1,2-diphenylethyne, complete conversion of **3a** to **Z-3b** was achieved within 5 s, followed by subsequent *Z/E* isomerization (Fig. 4d). But the presence of 1,2-diphenylethyne slowed down the hydroboration procedure, leading to a significantly longer reaction time of 120 s for the full conversion of **3a** to **Z-3b**. Notably, already a small amount of 1,2-diphenylethyne strongly inhibited the isomerization process. Even after 24 h, **Z-3b** was still observed in high yield and high *Z/E* ratio. This inhibition of the *Z/E* isomerization process can be attributed to the higher affinity of the alkyne (in this case 1,2-diphenylethyne) to the cobalt catalyst compared to **Z-3b**[56]. This phenomenon partially explains the unique kinetic behavior of the reaction and provides a rationale for the excellent *Z* selectivity, indicating that the *E* isomer can only form after nearly complete consumption of the alkyne

substrate (Fig. 4b, d). The influence of the alkyne was further investigated using a combination of the terminal alkynes phenylacetylene (**1a**) and 4-chlorophenylacetylene (**9a**). Separately, **9a** exhibited lower reactivity compared to **1a**. However, when they were mixed together, a reversal in the transformation rates of the two alkynes was observed, with **9a** undergoing a more rapid transformation (Fig. 5c and Supplementary Fig. 12). This observation suggests that the reaction may be influenced by the acidity of the alkynyl hydrogen in the alkyne substrate (although the KIE experiment indicated that activation of the sp C-H bond is not the rate-determining step; Fig. 5a, eq 3). In comparison to **9a**, phenylacetylene possesses a lower C(sp)-H acidity, resulting in inferior reactivity in the mixed system.

Based on the aforementioned experiments and relevant studies in the literature, a mechanistic pathway consistent with the observed results can be proposed. It is postulated that an in situ formed [Co(II)(CNC-$^i$Pr)] (CNC-$^i$Pr = 2,6-bis(3-isopropylimidazol-2-ylidene)pyridine) complex serves as the catalyst precursor. Supporting evidence for the formation of this complex (**Co-A**, see Fig. 5d and Supplementary Fig. 13) was obtained by analyzing the high-resolution mass spectrometry (HRMS) data from a solution containing CoBr$_2$, CNC-$^i$Pr, and KO$^t$Bu. Unfortunately, we were not able to isolate any stable [Co(II)(CNC-$^i$Pr)] complex because these undergo rather rapidly a disproportionation reaction. As a result, a single crystal of a [Co(III)(CNC-$^i$Pr)$_2$]$^{3+}$ complex (**Co-B**) was obtained (see Fig. 5e, Supplementary Figs. 14 and 15 and Supplementary Tables 13–15). An X-ray diffraction (XRD) study using a single crystal of **Co-B** confirms, however, the suggested coordination mode of the ligand and double deprotonation of the bis(imidazolium salt) CNC-$^i$Pr. The [Co(III)(CNC-$^i$Pr)$_2$]$^{3+}$ complex is presumed to represent the resting state of the active catalyst and using **Co-B** as catalyst precursor likewise results in significant catalytic activity (see Supplementary Fig. 16).

Upon reduction of the Co(II) precursor (Fig. 5f), an active Co-H species (**I**) was formed, which was observed by HRMS (Supplementary Fig. 17). Similar Co(I) species have been proposed in numerous hydrogenation[56–58] and hydroboration[38,59–62] reactions, and can be readily generated from Co(II)[38,56–61,63] or Co(III)[64] pincer complexes with suitable reductants. Subsequently, a reaction between **I** and a terminal alkyne leads to the deliberation of dihydrogen (H$_2$ was detected in course of the reaction, Supplementary Fig. 18) resulting in the formation of an alkynylcobalt complex (**II**). This complex then reacts with pinacolborane under oxidative addition of the B-H bond to yield an intermediate Co(III) species (**III**) which rearranges under reductive elimination to a Co(I) alkynylboronate complex (**IV**). Sequentially, a stereoselective *syn*-hydrocobaltation takes place, generating a vinyl-cobalt intermediate (**V**). After protonation and ligand exchange with a terminal alkyne, **V** ultimately leads to the formation of the *Z*-vinyl-boronate ester product and the regeneration of the alkynylcobalt complex **II**. When the alkyne substrate is (nearly) fully consumed, an insertion of the *Z*-vinylboronate ester into the Co-H bond may occur resulting in the generation of the corresponding *E*-vinylboronate ester through a *β*-H elimination step[56,60,61,65,66]. The mechanism is in accord with the labeling experiments and the different hydrogen atoms transferred during the reaction are visualized in Fig. 5f in blue, red, and black.

In summary, we have developed a highly efficient method for the stereoselective hydroboration of terminal alkynes, providing access to a wide range of vinylboronate ester compounds with excellent stereoselectivity. This transformation is achieved using a simple and cost-effective cobalt catalyst system generated in situ from a readily available CNC pincer ligand and Co(acac)$_2$. The catalyst system can be applied to a wide range of substrates and is especially suited for the conversion of terminal alkynes with sterically demanding substituents using commercial pinacol borane. This synthetic protocol exhibits excellent scalability, making it suitable for large-scale reactions without compromising neither the efficiency nor the selectivity. Impressive

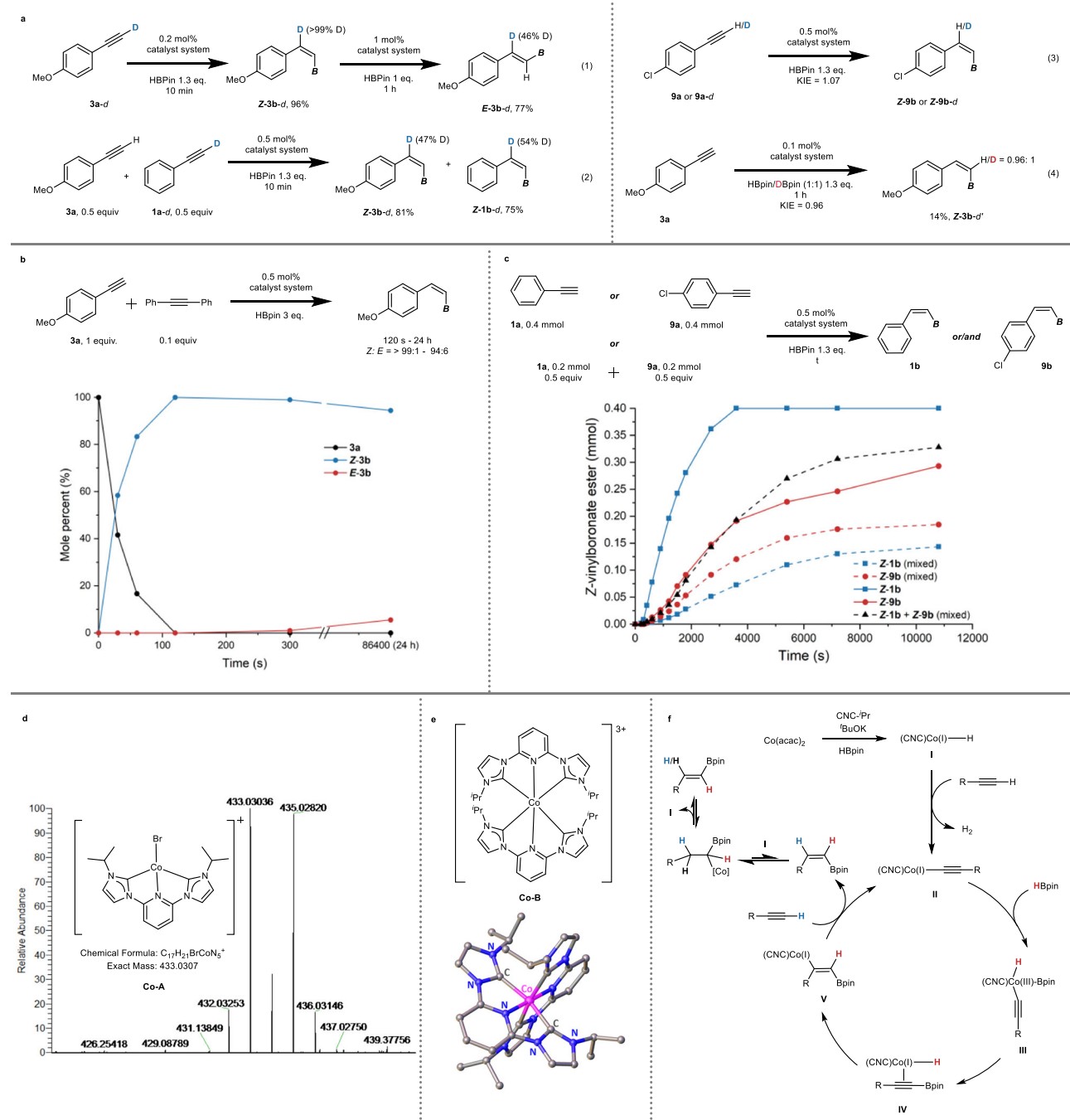

**Fig. 5 | Mechanistic study. a** Deuterium labeling experiments. **b** Inhibition of *Z/E* isomerization with an inert alkyne. **c** Kinetic trace of *Z*-selective products for reactions employing **1a**, **9a**, and mixture of **1a** and **9a**, respectively. Products of the mixed system have been marked in the profile. Reaction conditions: terminal alkyne (1.0 equiv, 0.4 mmol), HBpin (1.3 equiv), Co(acac)₂ ([Co], 0.5 mol%), CNC-*i*Pr (1.4 equiv to [Co]), *t*BuOK (5.6 equiv to [Co]) in DMF (0.5 ml) at room temperature (r.t.). **d** High Resolution Mass Spectrometry (HRMS) of [Co(II)(CNC-*i*Pr)Br]⁺. **e** X-ray structure of [Co(III)(CNC-*i*Pr)₂]Br₃. Hydrogen, bromine, and solvent atoms are omitted for clarity. See Supplementary Fig. 14 and Supplementary Tables 13–15 for details. **f** Proposed mechanism. The positions of the colored hydrogen atoms have been traced by the deuterium labeling experiments (**a**).

TONs and TOFs were achieved, with the highest hydroboration rate reported so far for the production of *Z*-vinylboronate esters. Kinetic investigations allow to propose a mechanism in which alkynes exhibit a higher affinity toward the cobalt catalyst compared to the *Z*-vinylboronate ester products. This preferential binding inhibits a fast *Z/E* isomerization process, which can only occur after almost complete consumption of the alkyne substrate to *Z*-vinylboronate ester. It is this special feature of the catalytic system that leads to a time-dependent stereoselectivity.

## Methods

### General procedure for catalytic Z-selective hydroboration

In a nitrogen atmosphere, a vail was charged with Co(acac)₂ (4.1 mg, 0.016 mmol), CNC-*i*Pr (10.2 mg, 0.022 mmol), *t*BuOK (10.1 mg, 0.09 mmol) in dry DMF (2 ml) and was stirred for 5 min. The freshly prepared stock solution of the in situ prepared active catalyst ([Co] = 8.0 mM in DMF) was added via a micro-syringe (50–500 µl, 0.0004–0.004 mmol, 0.1–1 mol%, as noted) to a vial charged with HBpin (75 µl, 0.52 mmol, 1.3 equiv., unless otherwise noted) and

DMF([alkyne] = 0.8 M). The mixture was stirred for 5 min. Alkyne (0.4 mmol, unless otherwise noted) was added rapidly and the resulting mixture was stirred for 12 h. The reaction was then quenched by adding water. Twenty ml EtOAc was added and the organic phase was washed with 10 ml brine twice to remove most of DMF. Pure product was isolated by column chromatography over silica gel deactivated with 2% NEt3 in petroleum ether using petroleum/EtOAc as the eluent. For some specific substrates, the reaction conditions were slightly changed, the detailed information of which can be found in the SI.

### General procedure for large scale catalytic Z-selective hydroboration of terminal alkynes

In nitrogen atmosphere, a 100 ml Schlenk vial was charged with alkyne (30 mmol) and dry DMF (20 ml), the alkyne solution was placed in an ice path. In another vial charged with $Co(acac)_2$ (38.6 mg, 0.15 mmol), CNC-$^i$Pr (96.0 mg, 0.21 mmol), $^t$BuOK (94.3 mg, 0.84 mmol), dry DMF (15 ml) was stirred for 5 min and HBpin (5.6 ml, 39 mmol) was added and stirred for a further 5 min. The catalyst solution was added dropwise into the alkyne solution under ice bath during 10 min. After that, ice bath was removed and the mixture was stirred at room temperature for a certain time. The reaction was then quenched by adding water. Three hundred ml EtOAc was added and the organic phase was washed with 3*100 ml water and 2*100 ml brine. the organic phase was passed through a short pad of silica gel. Pure product was obtained after removing volatiles under reduced pressure.

## Data availability

The data supporting the findings of this study are available within the paper and its Supplementary Information. The X-ray crystallographic coordinate for the structure reported (**Co-B**) has been deposited at the Cambridge Crystallographic Data Centre (CCDC), under deposition number CCDC 2288260. The data can be obtained free of charge from The Cambridge Crystallographic Data Centre via www.ccdc.cam.ac.uk. All data are available from the corresponding author upon request.

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

## Acknowledgements

P.H. thank Guangdong Provincial Department of Science and Technology (nos. 2019QN01L151, 2023A0505050137, 2022A1515011215), Guangzhou Municipal Science and Technology Bureau (no. 202201011797), GBRCE for Functional Molecular Engineering, Sun Yat-sen University, and State Key Laboratory of Structural Chemistry (FJIRSM, CAS) for support. We would like to express our gratitude to Prof. David Milstein (Weizmann Institute of Science) for reviewing the article and providing valuable suggestions.

## Author contributions

P.H. designed and directed the project. J.W. carried out catalytic experiments. Y.H. and Y.Z. discussed and shared ideas regarding the design of the experiments. J.W., H.G., and P.H. wrote the manuscript. All authors discussed the results and commented on the manuscript.

## Competing interests

The authors declare no competing interests.
