## [Peer Review File · Nature Communications]

Cobalt catalyzed practical hydroboration of terminal alkynes with time-dependent stereoselectivityREVIEWER COMMENTS

Reviewer #1 (Remarks to the Author):

The manuscript by Peng and co-workers (manuscript ID: NCOMMS-23-44051) describes the efficient synthesis of Z-vinylboronates esters via an in-situ formed cobalt(I) CNC pincer complex. Overall, the manuscript is well written and the kinetic and deuterium labeling experiments have been performed with care and seem to support their mechanistic proposal. Their reported method is a significant improvement over the state of the art as reported in their introduction. Furthermore, their methodology tolerates a wide variety of functional groups, supports a wide variety of substitution patterns at the phenyl acetylene, and is even Z-selective for linear aliphatic alkynes. However, the reported methodology is far from general; key mechanistic factors have not been explained, and there seem to be reproducibility issues with certain experiments (see my comments below). As such I am not able to accept the manuscript in its current form. However, I am willing to review a heavily revised manuscript that also includes a computational mechanistic investigation.

Comments:

1. The authors claim that all the substrates have been synthesized according to General Procedure A. However, the procedure is far from general; between substrates the catalyst loading varies between (0.05 and 1.0 mol%) and the amount of HBpin varies between (1.0 and 3.0 equiv.). Furthermore, in contrast to what the caption of Figure 2 indicates, the substrate also varies between 0.2 – 0.8 mmol and sometimes even additives are added (e.g., for 7b 10 mol% diphenylacetylene was used) that are not mentioned. Therefore, it seems that for every substrate an optimization protocol needs to be performed that severely limits applicability.
2. Mechanistically there are two very important questions that are not explained: (i) the role of the HBpin and (ii) the role of the catalyst. As evident from Table S4, using 3 equiv. of HBpin for 12 hours results in almost exclusive E-selectivity (98:2), while using 2 equiv. of HBpin for 24 hours results in almost Z-selectivity (99:1), which needs to be explained. Likewise, for the catalyst loading, under identical conditions, it is evident from Table S9 (entries 4 and 11) that with 0.2 mol% Co(acac)₂ and 1.4 eq. of ligand the E-isomer is predominant (E/Z; 86:8), while with 0.1 mol% Co(acac)₂ and 1.4 eq. of ligand the Z-isomer is predominant (E/Z: 5:89). This needs to be explained. As it stands Just by changing the catalyst loading from 0.2 mol% to 0.1 mol% the stereoselectivity is completely reversed. How feasible is this methodology to run a wide scope? It seems like that for every substrate a big optimization procedure is required.
3. There are several discrepancy, and perhaps apparent reproducibility issues, in Tables S4-S10. To highlight:
 - (i) Table 1 (Entry 7; 4:96) does not match with Table S3 (Entry 4; 28:72). Same conditions, but different E/Z ratios.
 - (ii) Table S6 (Entry 9; trace and 44%) does not match with Table S7 (Entry 4; Trace and 14%). Same conditions different yields of the Z-isomer.
 - (iii) Table S8 (Entry 7; trace and 50%) and Table S9 (entry 9; trace and 8%). Same conditions different yields of the Z-isomer

(iv) Table S4; entries 3 and 4. How can the reaction yield go down after longer reaction times; are different products formed at prolonged times?

(v) In Table 1; Entry 9, the yield is described as 44% (I assume of both E/Z) with a ratio of 95:5. However, in Table S6 Entry 9 the yield of the Z isomer is described as 44%. In short; with a ratio of 95:5 and a 44% overall yield, the actual percentage of Z is 41.8 and that of E is 3.3%.

4. Neither in screening of the ligand (Table S9), nor that of the metal precursor (Table S8), it is evident that Co(acac)₂ is the best catalyst. Instead CoCl₂ seems to be best. How did the authors based on the data reach the conclusion that Co(acac)₂ is the best catalyst?

5. From the kinetic measurements in Figure 4b it is evident that the Z-product is formed in about 100-120 second (see also Figure 3D). How come that the reaction takes 12 hours to complete in the optimization protocols and General procedure A?

6. In Figure 5B and 5C, there seems to be no induction period, in contrast to what is reported in Figure 4B. Please explain.

7. The authors claim an induction period of 70s in which the active catalyst is formed, but this is highly substrate specific and by no means general as several products in Figure 3D are formed within 5 seconds (e.g., Z-2b and Z-3b). This difference justifies an explanation.

8. Figure 5a. how is it possible with 1.3 equiv. of HPin to get 150% combined yield for both alkynes?

9. The authors attempted the isolation of the catalyst precursor by using a 1:1.1 Co/Ligand ratio, but the actual reaction under the experimental reaction conditions is 1:1.4. This reaction should be checked and mentioned what the products of the reaction.

10. HRMS confirms the cationic Co(II) complex still structural evidence is lacking. The crystallization was performed in methanol (this might cause the disproportionation to form a Co(III) complex).

11. The putative Co-H has been synthesized with NaBHEt₃ but it is only supported with HRMS data, which assigns it as an anionic Co(0)-H, quite different than what is proposed in the reaction.

12. To further confirm their mechanism, DBpin should be used in their deuterium labelling experiments.

13. The manuscript would benefit from a computational investigation into the reaction mechanism in support of their experimental findings.

14. In figure S8, the integration of the peak at 5.5 ppm (0.49) does not seem to match the deuterium incorporation of 96% as mentioned.

Reviewer #2 (Remarks to the Author):

In this manuscript Hu and coworkers reported the Co catalyzed hydroboration of terminal alkynes. Depending on the reaction conditions, Z or E isomer could be synthesized in highly selective manner, and a readily available ligand was used. The scope and limitation as well as the mechanism of the reaction were examined appropriately.

Based on the high quality and strong impact of the study, I recommend this manuscript for publication Nature Communications. I suggest the authors to consider following issues before submitting the final version of the manuscript.

Page 3, line 100. "two terminal alkene substituents" should read "two ethynyl groups"

Reviewer #3 (Remarks to the Author):

Hu et al. report the synthesis of selective Z-1-vinyl boron compounds by the addition of a B-H bond to terminal alkynes using cobalt salt and a readily accessible air-stable CNC pincer ligand. The authors carried out the Kinetic studies which reveal a formal second-order dependence on cobalt concentration. A mechanistic investigation of the catalytic reaction indicates that the alkynes exhibit a higher affinity for the catalyst than the alkene products, resulting in exceptional Z-selective performance. The reviewer has thoroughly checked the manuscript. The work is very interesting. A good substrate scope and generality of the catalytic protocol are reported. The manuscript is nicely written. However, there are serious drawbacks are following:

1. In the introduction part the authors nicely mentioned the three pincer Co and Fe complexes have been developed to facilitate this transformation with favorable Z-selectivity (Fig. 1a). These catalysts are well defined. However, in the current work, the catalyst is not well defined. Have the authors isolated the actual cobalt complex out of the reaction between $\text{Co}(\text{acac})_2$ and a CNC-iPr ligand? What is the structure of this complex? During the reaction, Cobalt gets oxidized to from +2 to +3 state. A detailed understanding of the structure and bonding is required to define the catalyst.
2. While this work shows the use of a new cobalt catalytic system, but several other similar methods exist to generate a similar type of Z-selective hydroboration, but it is not clear what the advantage of this method is over previously published examples. Cobalt catalyst is already used for Z-selective hydroboration (Reference 37).
3. Authors reported that the longer reaction time was directed to the E isomer, but the exact reaction time and mechanism were not optimized. But it would be interesting to know how isomerization occurs and what is the role of catalyst.
4. They used base tBuOK 5.6 equiv. with respect to [Co] catalyst but what is the role of excess base is not clear in the mechanism.
5. Authors write "few catalytic reactions leading selectively to the Z-isomers have been reported". This statement is not correct. There are many reports.
6. Authors should cite the following article.

Yuqing Zhang, Yanhui Chen, Qingyu Tian, Binju Wang, Guolin Cheng. Palladium-Catalyzed Multicomponent Assembly of (Z)-Alkenylborons via Carbopalladation/Boronation/Retro-Diels–Alder Cascade Reaction. *The Journal of Organic Chemistry* 2023, 88 (16) , 11793-11800. <https://doi.org/10.1021/acs.joc.3c01084>

7. ¹¹B NMR of the products should be provided for all the organoboron compounds.
8. Have the authors used Catechol boron (BCat) or other boranes as the boron source?
9. In SI Fig. S2. is not clean. A better picture should be given.
10. The mechanism of the reaction is not very convincing.

Overall in the current manuscript, the novelty of the work is missing and a major revision is recommended for publication in *Nature Communications*.

Point-by-point responses to the comments of reviewers

Reviewer #1 (Remarks to the Author):

Comments: *The manuscript by Peng and co-workers (manuscript ID: NCOMMS-23-44051) describes the efficient synthesis of Z-vinylboronates esters via an in-situ formed cobalt(I) CNC pincer complex. Overall, the manuscript is well written and the kinetic and deuterium labeling experiments have been performed with care and seem to support their mechanistic proposal. Their reported method is a significant improvement over the state of the art as reported in their introduction. Furthermore, their methodology tolerates a wide variety of functional groups, supports a wide variety of substitution patterns at the phenyl acetylene, and is even Z-selective for linear aliphatic alkynes. However, the reported methodology is far from general; key mechanistic factors have not been explained, and there seem to be reproducibility issues with certain experiments (see my comments below). As such I am not able to accept the manuscript in its current form. However, I am willing to review a heavily revised manuscript that also includes a computational mechanistic investigation.*

Response: We are very grateful for the helpful suggestions of the reviewer, whose comments have greatly improved the quality of the article after revision. For a detailed response to these comments, please see the point-by-point replies below. We hope that our responses and revisions will satisfy the reviewers and better showcase this method.

Comments: *1. The authors claim that all the substrates have been synthesized according to General Procedure A. However, the procedure is far from general; between substrates the catalyst loading varies between (0.05 and 1.0 mol%) and the amount of HBpin varies between (1.0 and 3.0 equiv.). Furthermore, in contrast to what the caption of Figure 2 indicates, the substrate also varies between 0.2 – 0.8 mmol and sometimes even additives are added (e.g., for 7b 10 mol% diphenylacetylene was used) that are not mentioned. Therefore, it seems that for every substrate an optimization protocol needs to be performed that severely limits applicability.*

Response: We apologize for any confusion caused by our unclear description. We have made revisions to Figure 2 (including the note), general procedure A, and thoroughly reviewed the experimental details to prevent further misunderstandings. In order to demonstrate the generality of the method, new reactions have been carried out using the general procedure, and the results have been included in the revised Figure 2 and the Supporting Information. The results are also listed below.

The diverse conditions presented are primarily due to the high activity of the catalyst system. In an effort to showcase the system's activity, we aimed to minimize catalyst loading while maintaining high yield and selectivity, resulting in a wide range of conditions. In fact, most aryl alkynes (**1a**, **3a-7a**, **13a**, **14a**, **17a**, **18a**, **22a-30a**) can undergo Z-selective hydroboration under conditions employing 0.5 mol% catalyst loading, while aryl alkynes bearing electron-withdrawing groups (**8a-12a**, **15a**, **16a**, **19a-21a**) and other alkynes (**31a-42a**) required a higher catalyst loading of 1.0 mol%

to achieve satisfactory yields of the desired Z-products. Therefore, either 0.5 mol% or 1 mol% catalyst loading is sufficient to catalyze the reaction for all tested substrates, depending on the reactivity of the substrate.

For substrates that require multistep synthesis or are expensive, we conducted the reactions on a 0.2 mmol scale to minimize costs. Additionally, in order to demonstrate the high activity of the catalyst system, we performed reactions with low catalyst loading (**1a**, **3a**). When using low catalyst loading (0.1 mol% or less), we employed an 0.8 mmol substrate to reduce potential errors during experimental handling (such as catalyst addition). These substrates exhibited no issues with 0.4 mmol scale reactions.

Figure below illustrates several examples of reactions carried out under general conditions A. Substrates **1b**, **3b**, **13b** and **18b** displayed good reactivity with low catalyst loadings also yielded similar outcomes with 0.5 mol% catalyst.

In addition, under 0.5 mol% catalyst loading, moderate yields of the desired products can also be obtained for certain substrates with relatively lower reactivity (**4b**, **7b**, **9b**, **10b**, **14b**, **21b**, **29b**). Increasing the catalyst loading from 0.5 mol% to 1 mol% significantly enhances the yields. In the case of certain less reactive substrates (**29b**), a higher amount of HBpin can lead to a higher yield while maintaining the same Z-selectivity. However, substrate **12b** was an exception where both isomerization and hydroboration occurred simultaneously. By adding 10 mol% of diphenylacetylene, a product with high Z-selectivity could be obtained, albeit with a reduced yield. The corresponding notes have been added in the revised manuscript.

Note: the data below products are catalyst loading (0.1-1mol%), amount of HBpin (1.3-3 eq.), yield and ratio of Z/E. See revised manuscript and Supplementary Information for experimental details.

Comments: 2. Mechanistically there are two very important questions that are not explained: (i)

the role of the HBpin and (ii) the role of the catalyst. As evident from Table S4, using 3 equiv. of HBpin for 12 hours results in almost exclusive *E*-selectivity (98:2), while using 2 equiv. of HBpin for 24 hours results in almost *Z*-selectivity (99:1), which needs to be explained. Likewise, for the catalyst loading, under identical conditions, it is evident from Table S9 (entries 4 and 11) that with 0.2 mol% Co(acac)₂ and 1.4 eq. of ligand the *E*-isomer is predominant (*E/Z*: 86:8), while with 0.1 mol% Co(acac)₂ and 1.4 eq. of ligand the *Z*-isomer is predominant (*E/Z*: 5:89). This needs to be explained. As it stands Just by changing the catalyst loading from 0.2 mol% to 0.1 mol% the stereoselectivity is completely reversed. How feasible is this methodology to run a wide scope? It seems like that for every substrate a big optimization procedure is required.

Response: We're sorry for the confusion caused by the excessive data. The observed phenomenon, where seemingly minor changes in reaction conditions result in significant variations in the reaction outcome, is attributed to the high activity and the unique catalytic process of our system: the complete formation of the *Z*-alkene from the substrate, followed by its subsequent conversion to the *E*-alkene. The generation of the *Z*-structure is completed before the *E*-structure begins to form through isomerization. This phenomenon is general for various substrates and is the fundamental reason why the reaction can be used to prepare a wide variety of *Z*-alkenes in high *Z/E* ratio. This process has been thoroughly demonstrated and discussed in the kinetic study section of our paper (see Fig. 4 and the corresponding discussion, see also the selected mechanism study part below). In contrast to previously reported catalytic systems, our catalytic system exhibits an unusual kinetic response, particularly with respect to the factors of HBpin and catalyst. Increasing the amounts of HBpin and catalyst (or using a more effective catalyst, such as Co(acac)₂, which is more active than CoCl₂) significantly accelerates the reaction process. It is based on this behavior that the phenomenon mentioned by the reviewers arises: high levels of HBpin and catalyst (or using the more effective catalyst Co(acac)₂) can lead to a complete reversal of the reaction's *Z/E* selectivity. For our catalytic system, this is simply a normal consequence of the kinetic effects.

Fig 4b Kinetic profile of the hydroboration of phenylacetylene

To better understand this process, please refer to Table S3, Table S4, and Table S8. Table S3 illustrates the transition of alkene products from the *Z* configuration to the *E* configuration as the reaction time increases. Table S4 demonstrates that increasing the amount of HBpin can accelerate the reaction, resulting in a higher proportion of *E*-configured products within the same reaction time. Conversely, to obtain a higher proportion of *Z*-configured products, a smaller amount of HBpin can be used within a shorter time frame. Table S8 shows that Co(acac)₂ is the most active catalyst, as it leads to higher conversion rates and a greater proportion of *E*-configured products compared to the other cobalt salts when used for the same duration and at the same catalyst loading.

Concerns about the applicability of optimized conditions and the need for different optimizations for different substrates have been addressed in our previous response. We hope that the explanation provided above regarding the reaction process will further elucidate this catalytic system for the reviewer. In fact, due to the presence of three sensitive and stable forward kinetic response factors in our reaction system (time, catalyst, and HBpin dosage, with temperature fixed at room temperature and not under consideration), longer reaction times, increased catalyst and HBpin dosages all promote the forward progression of the reaction (resulting in the generation of *Z*-alkenes from alkynes, which further convert to *E*-alkenes). Therefore, our reaction encompasses a broad range of optimization conditions. By adjusting these three forward factors, many high-yield and selective reaction conditions can be obtained, rather than being restricted to a single limiting optimization condition, which sets it apart from many common catalytic systems and may be a cause for the reviewer's perplexity. For instance, substrate **3a**, under a wide range of conditions, consistently leads to good yields and high *Z*-alkene selectivity (please see Fig. 2, 3, 4d, 5b). If general reaction conditions for different substrates are required, moderate reaction factors suffice, such as 0.5 mol% or 1 mol% catalyst loading and 1.3 equivalents of HBpin, making the majority of substrates applicable. Even if the general optimization conditions are not suitable for a particular substrate, specific optimization conditions can be quickly obtained by adjusting the above-mentioned three factors (as all three factors are positive promoters of the reaction process, which is consistent for all substrates). The reviewer can confirm this point from the different reaction results listed in our manuscript under various conditions.

Comments: 3. *There are several discrepancy, and perhaps apparent reproducibility issues, in Tables S4-S10. To highlight:*

(i) Table 1 (Entry 7; 4:96) does not match with Table S3 (Entry 4; 28:72). Same conditions, but different E/Z ratios.

(ii) Table S6 (Entry 9; trace and 44%) does not match with Table S7 (Entry 4; Trace and 14%). Same conditions different yields of the Z-isomer.

(iii) Table S8 (Entry 7; trace and 50%) and Table S9 (entry 9; trace and 8%). Same conditions different yields of the Z-isomer

(iv) Table S4; entries 3 and 4. How can the reaction yield go down after longer reaction times; are different products formed at prolonged times?

(v) In Table 1; Entry 9, the yield is described as 44% (I assume of both E/Z) with a ratio of 95:5. However, in Table S6 Entry 9 the yield of the Z isomer is describes as 44%. In short; with a ratio of 95:5 and a 44% overall yield, the actual percentage of Z is 41.8 and that of E is 3.3%.

Response: We thank the reviewer for the suggestions. We find that the discrepancies pointed out are not issues of reproducibility, but rather mistakes. During the preliminary stages of our investigation, we conducted numerous reactions to refine the reaction conditions. When incorporating these conditions into the Supplementary Information, some data was erroneously recorded. We deeply regret the presentation of inaccurate data in the article and have taken steps to double-check and revise any incorrect data.

Corrections and point-by-point reply are listed below:

(i) Table 1 (Entry 7; 4:96) does not match with Table S3 (Entry 4; 28:72). Same conditions, but different E/Z ratios.

The data of Table S3 entry 4 was mistakenly entered and has been revised as below.

Table S3 Initial screening of time

Entry	t	Yield (E -1b: Z -1b)
1	8 h	>99% (95:5)
2	6 h	99% (95:5)
3	30 min	>99% (51:49)
4	10 min	>99% (4:96)

Condition: 5 mol% CoCl₂, 5 mol% CNC-iPr, 20 mol% ^tBuOK, Solvent: DMF, [1a]=0.2 M, 3 eq. HBpin, r.t.

(ii) Table S6 (Entry 9; trace and 44%) does not match with Table S7 (Entry 4; Trace and 14%). Same conditions different yields of the Z-isomer.

The data of Table S7 entry 4 was mistakenly entered and has been revised as below.

Table S7 Screening of solvent

Entry	Metal	Solvent	E -1b	Z -1b
1	0.2 mol% CoCl ₂	DMA	Trace	9%

2	0.2 mol% CoCl ₂	NMP	Trace	6%
3	0.2 mol% CoCl ₂	DMSO	Trace	Trace
4	0.2 mol% CoCl ₂	DMF	Trace	44%
5	0.3 mol% CoCl ₂	DMA	Trace	17%
6	0.3 mol% CoCl ₂	NMP	Trace	13%
7	0.3 mol% CoCl ₂	DMSO	Trace	Trace
8	0.3 mol% CoCl ₂	DMF	Trace	58%

Conditions: x mol% CoCl₂, x mol% **CNC-ⁱPr**, 4x mol% ^tBuOK, Solvent, [**1a**] = 0.4 M, 3 eq. HBpin, t = 12 h, 25 °C.

(iii) Table S8 (Entry 7; trace and 50%) and Table S9 (entry 9; trace and 8%). Same conditions different yields of the Z-isomer

The conditions of Table S8 were mistakenly entered and have been revised as below.

Table S8 Screening of Metal Precursor

Entry	Metal	E -1b	Z -1b
1	0.3 mol% CoCl ₂	Trace	91%
2	0.2 mol% CoCl ₂	Trace	27%
3	0.4 mol% CoBr ₂	Trace	48%
4	0.3 mol% CoBr ₂	Trace	35%
5	0.3 mol% Co(acac) ₂	52%	46%
6	0.2 mol% Co(acac) ₂	48%	50%
7	0.1 mol% Co(acac) ₂	Trace	50%
8	0.05 mol% Co(acac) ₂	Trace	Trace
9	0.3 mol% Co(OAc) ₂	50%	52%
10	0.2 mol% Co(OAc) ₂	14%	83%
11	0.1 mol% Co(OAc) ₂	Trace	45%
12	0.05 mol% Co(OAc) ₂	Trace	12%

Conditions: x mol% Metal, 1.1x mol% **CNC-ⁱPr**, 4x mol% ^tBuOK, Solvent: DMF, [**1a**] = 0.4 M, 3 eq. HBpin, t = 12 h, 25 °C.

(iv) Table S4; entries 3 and 4. How can the reaction yield go down after longer reaction times; are different products formed at prolonged times?

The data of Table S4 entry 4 was mistakenly entered and has been revised as below.

Table S4 Initial screening of equivalent of HBpin

Entry	HBPiN	t	Yield (E -1b: Z -1b)
1	1.5 eq.	10 min	65% (<1:99)
2	1.5 eq.	24 h	70% (<1:99)
3	2 eq.	10 min	87% (<1:99)
4	2 eq.	24 h	95% (66:34)
5	3 eq.	10 min	>99% (4:96)
6	3 eq.	12 h	99% (98:2)

Conditions: 5 mol% CoCl₂, 5 mol% CNC-ⁱPr, 20 mol% ^tBuOK, Solvent: DMF, [1a] = 0.2 M, HBPiN, r.t.

(v) In Table 1; Entry 9, the yield is described as 44% (I assume of both *E/Z*) with a ratio of 95:5. However, in Table S6 Entry 9 the yield of the *Z* isomer is describes as 44%. In short; with a ratio of 95:5 and a 44% overall yield, the actual percentage of *Z* is 41.8 and that of *E* is 3.3%.

We have reviewed the data in question and found that in both Table 1, Entry 9 and Table S6, Entry 9, the yield of the *Z*-isomer was 44%, with a trace signal of the *E* product (after rectification, *Z/E*=97:3, based on crude ¹H-NMR). We have revised Table 1, Entry 9 to include the yield of the *E*-isomer. Additionally, we have revised the conditions for note e (the ligand should be 0.22 mol%). Table S6, entry 2 has also been revised (*Z/E*=96:4 after rectification).

Crude ¹H-NMR of table 1, entry 9 with mesitylene as internal standard

Table 1. Optimization study

entry	precursor	ligand	yield (%) ^a	Z/E ratio ^b
1	CoCl ₂	CNC-H	0	-
2	CoCl ₂	CNC-Me	71%	3:97
3	CoCl ₂	CNC-Et	78%	36:64
4	CoCl ₂	CNC-ⁱPr	99%	2:98
5	CoCl ₂	CNC-^tBu	7%	<1:99
6	CoCl ₂	CNC-Mes	0	-
7 ^c	CoCl ₂	CNC-ⁱPr	99%	96:4
8 ^d	CoCl ₂	CNC-ⁱPr	70%	>99:1
9^e	CoCl₂	CNC-ⁱPr	45%	97:3
10 ^e	Co(acac) ₂	CNC-ⁱPr	98%	51:49
11 ^f	Co(acac) ₂	CNC-ⁱPr	99% (90%) ^g	98:2
12 ^{f,h}	Co(acac) ₂	CNC-ⁱPr	98% (90%) ^g	>99:1

Reaction conditions: CoCl₂ (5.0 mol%), **1a** (0.4 mmol, 1.0 equiv), ligand (5.0 mol%), ^tBuOK (20.0 mol%), HBpin (3.0 equiv), DMF (2 mL), room temperature (r.t.), N₂ atmosphere, 12 h. ^aNMR yield determined using methylene bromide as the internal standard. ^bZ/E ratio based on ¹H NMR analysis. ^cReacted for 10 min. ^dHBpin used in 1.5 equiv., **24 h**. ^eCo precursor (0.2 mol%), **ligand (0.22 mol%)**, ^tBuOK (0.8 mol%), DMF (1 mL). ^fCo(acac)₂ (0.1 mol%), ligand (0.14 mol%), ^tBuOK (0.56 mol%), DMF (0.5 mL). ^gIsolated yield. ^h**HBpin (1.3 equiv.)**.

Table S6 Screening of reaction concentration

Entry	Metal	[1a]	E-1b	Z-1b
1	0.4 mol% CoCl ₂	0.05 M	Trace	7%
2	0.4 mol% CoCl ₂	0.1 M	4%	96%
3	0.4 mol% CoCl ₂	0.2 M	76%	24%
4	0.4 mol% CoCl ₂	0.4 M	97%	4%
5	0.2 mol% CoCl ₂	0.05 M	Trace	Trace
6	0.2 mol% CoCl ₂	0.1 M	Trace	Trace
7	0.2 mol% CoCl ₂	0.2 M	Trace	8%
8	0.2 mol% CoCl ₂	0.3 M	Trace	22%
9	0.2 mol% CoCl ₂	0.4 M	Trace	44%

10 ^a	0.5mol% CoCl ₂	neat	Trace	Trace
11 ^b	0.5 mol% CoCl ₂	neat	Trace	Trace

Conditions: x mol% CoCl₂, x mol% **CNC-ⁱPr**, 4x mol% **^tBuOK**, Solvent: DMF, 3 eq. HBpin, t= 12 h, 25 °C. a: 200 µl of DMF to dissolve the Cat., 4 mmol scale; b: 200 µl of DMF to dissolve the Cat., 10 mmol scale.

Comments: 4. *Neither in screening of the ligand (Table S9), nor that of the metal precursor (Table S8), it is evident that Co(acac)₂ is the best catalyst. Instead CoCl₂ seems to best. How did the authors based on the data reach the conclusion that Co(acac)₂ is the best catalyst?*

Response: As previously discussed, we found the formation of the *Z* product occurs initially and continues until the alkyne substrate is completely consumed. Subsequently, the *Z*-alkene product undergoes an isomerization process, resulting in the formation of the *E*-alkene. Acting as a catalyst, Co(acac)₂, as shown in Table S8, initially converts all alkynes into *Z* products and also facilitates the conversion of approximately half of the *Z* products into *E* products. In contrast, under identical catalytic conditions, CoCl₂ exclusively generates the *Z*-type product without fully consuming the substrate. Hence, it can be deduced that Co(acac)₂ exhibits significantly higher catalytic activity compared to CoCl₂ (refer also to Table 1, entry 9 vs. 11). The *Z/E* selectivity can be adjusted by varying the reaction time or the catalyst loading when utilizing Co(acac)₂.

Comments: 5. *From the kinetic measurements in Figure 4b it is evident that the Z-product is formed in about 100-120 second (see also Figure 3D). How come that the reaction takes 12hours to complete in the optimization protocols and General procedure A?*

Response: The reaction conditions utilized in Figure 4b (2 equiv. of HBpin for **1a**) and general condition A (1.3 equiv. of HBpin for **1a**) are different. As previously mentioned, by manipulating three key factors (reaction time, HBpin dosage, and catalyst loading), numerous high-yield and selective reaction conditions can be achieved. In Figure 4b, we opted for conditions that facilitate rapid reaction to enable a convenient kinetic data collection. Furthermore, a larger quantity of HBpin has a beneficial effect on enhancing the *Z/E* transformation process, which is slower in comparison to the formation of the *Z*-isomer. Conversely, a 12-hour reaction time is suitable for a broad substrate scope and was selected for general condition A.

Comments: 6. *In Figure 5B and 5C, there seems to be no induction period, in contrast to what is reported in Figure 4B. Please explain.*

Response: The substrate 4-methoxyphenylacetylene (**3a**) exhibits a remarkably high reactivity, leading to the rapid occurrence of hydroboration. (Under identical conditions as in fig 5b but without the addition of the inert alkyne, complete conversion of the substrate occurs within only 5 s). The first aliquot was taken at 30 s. However, by that time, the reaction was already halfway complete. Thus, the induction period cannot be recorded.

For Figure 5c, after zoom out, induction periods lasting for 200 seconds or longer will be found. (see Figure below). The Figure below has been added in the revised SI as Fig. S12.

Fig. S12. The first 600 seconds of Fig 5c

Comments: 7. The authors claim an induction period of 70s in which the active catalyst is formed, but this is highly substrate specific and by no means general as several products in Figure 3D are formed within 5 seconds (e.g., Z-2b and Z-3b). This difference justifies an explanation.

Response: As depicted in the above Figure S12, the presence of an induction period is a common occurrence. However, the duration of the induction period varies depending on the specific reaction conditions and substrate. This discrepancy arises from the differing concentrations of the actual active catalyst under varying reaction conditions. Additionally, distinct substrates necessitate different catalyst concentrations to initiate reactions due to their varying reactivity. It is noteworthy that in the case of the electron-rich alkyne **3b**, the reaction progresses at an exceptionally rapid rate, requiring only a minimal amount of active catalyst to be generated during the induction period in order to initiate the high-speed transformation.

Comments: 8. Figure 5a. how is it possible with 1.3 equiv. of HPin to get 150% combined yield for both alkynes?

Response: The description in Figure 5a is inaccurate and has been revised. In reality, 0.5 equiv. of each alkyne were added to the reaction. We genuinely appreciate the reviewer for highlighting these particular errors.

Comments: 9. The authors attempted the isolation of the catalyst precursor by using a 1:1.1 Co/Ligand ratio, but the actual reaction under the experimental reaction conditions is 1:1.4. This reaction should be checked and mentioned what the products of the reaction.

Response: Please refer to revised Table S8, all the reactions were performed with a 1:1.1 Co/Ligand ratio. The same reaction and similar results were observed.

Comments: 10. HRMS confirms the cationic Co(II) complex still structural evidence is lacking. The crystallization was performed in methanol (this might cause the disproportionation to form a Co(III) complex).

Response: We have been engaged in endeavors to grow single crystals for a period of two years, conducting hundreds of attempts. Our efforts involved the exploration of various solvents, including MeOH, Tol, Et₂O, CH₃CN, EA, DCM, Hexane, Pentane, etc. Additionally, auxiliary ligands such as pyridine, bipyridyl, and PPh₃ were incorporated to facilitate the growth of single crystals. Unfortunately, none of these strategies led to successful crystallization, likely due to the high reactivity of the catalyst. Furthermore, in addition to the previously mentioned Et₂O/MeOH combination (SI, section 6.8), alternative solvent systems such as EA/DMF or EA/DMSO also facilitated the formation of Co(III) complex crystals. This information has been included in the revised SI. Moreover, the Co(III) complex was employed as a catalyst for the reaction, yielding similar outcomes. This finding further corroborates the structural information of the cationic Co(II) complex.

Comments: 11. The putative Co-H has been synthesized with NaBHET₃ but it is only supported with HRMS data, which assigns it as an anionic Co(0)-H, quite different than what is proposed in the reaction.

Response: Efforts to obtain single crystals of Co-H were unsuccessful despite numerous attempts. The presence of the Co-H species was confirmed via HRMS, which indicated the presence of an electroneutral Co(I)-H species rather than an anionic Co(0)-H complex, as evidenced by the observation of the [M+Na]⁺ peak under positive ESI-source (anionic Co(0)-H complex cannot give the peak under positive ESI-source).

Comments: 12. To further confirm their mechanism, DBPin should be used in their deuterium labelling experiments.

Response: Deuterium labeling experiments using DBPin were conducted to further validate the reaction mechanism. The results have been incorporated into the revised SI (also refer to the figure below) as Fig. S5. The hydroboration of 1-(4-methoxyphenyl) acetylene (**3a**) yielded *Z*-**3b-d'** with a notable deuterium retention of 96% at the alpha-position of the Bpin group, consistent with the deuterium labeling experiments in Fig. 5a, eq 1. When employing 1 mol% Co(acac)₂/CNC-iPr and 1 equiv of DBPin, isomerization of *Z*-**3b-d'** to *E*-**3b-d'** occurred, resulting in a significant deuterium incorporation rate of 32% at the beta-position of the Bpin group, similar to the reaction using HBPin (Fig. 5a, eq 1), which exhibited a reduced deuterium percentage.

Fig. S5 Deuterium labeling experiments with DBpin

Comments: 13. *The manuscript would benefit from a computational investigation into the reaction mechanism in support of their experimental findings.*

Response: We agree with the reviewer's perspective that incorporating computational results may be beneficial for understanding the mechanism, though based on our kinetic experiments, mechanistic studies, and previously published related research, the proposed mechanism is already reasonable. Nevertheless, after consulting several computational chemistry experts, we have learned that each of them has a long waiting list for obtaining computational results, and the earliest collaboration we could commence would require a six-month waiting period, not including the time to get the final computational results. Therefore, at present, it is not practical to include computational results in this work. Given the substantial volume of data in this study, we plan to undertake a dedicated collaboration with computational chemistry experts in a subsequent project to conduct thorough and comprehensive computational research.

Comments: 14. *In figure S8, the integration of the peak at 5.5 ppm (0.49) does not seem to match the deuterium incorporation of 96% as mentioned.*

Response: The kinetic isotope effect (KIE) experiment of HBpin/DBpin shown in Figure S8 (H/D=0.96) does not indicate a deuterium incorporation of 96%. Rather, it signifies the ratio of H atoms to D atoms, calculated as " $H/D=0.49/(1-0.49)=0.96$ ". To avoid misunderstandings, revisions have been made in Figure 5a, eq4 and SI.

In summary, we are very grateful for the constructive suggestions provided by the reviewer, which have significantly enhanced the quality of our work. We have expressed our gratitude to the reviewer in the acknowledgments section. We regret that we are currently unable to provide computational chemistry results, and we hope that our future collaborative research with computational chemists will further enhance the understanding of the mechanism.

Reviewer #2 (Remarks to the Author):

Comments: *In this manuscript Hu and coworkers reported the Co catalyzed hydroboration of terminal alkynes. Depending on the reaction conditions, Z or E isomer could be synthesized in highly selective manner, and a readily available ligand was used. The scope and limitation as well as the mechanism of the reaction were examined appropriately.*

Based on the high quality and strong impact of the study, I recommend this manuscript for publication Nature Communications. I suggest the authors to consider following issues before submitting the final version of the manuscript.

Page 3, line 100. "two terminal alkene substituents" should read "two ethynyl groups"

Response: We thank the reviewer for the positive comments and the suggestion. "two terminal

alkene substituents” has been revised to “two ethynyl groups”. We have expressed our gratitude to the reviewer in the acknowledgments section.

Reviewer #3 (Remarks to the Author):

Hu et al. report the synthesis of selective Z-1-vinyl boron compounds by the addition of a B-H bond to terminal alkynes using cobalt salt and a readily accessible air-stable CNC pincer ligand. The authors carried out the Kinetic studies which reveal a formal second-order dependence on cobalt concentration. A mechanistic investigation of the catalytic reaction indicates that the alkynes exhibit a higher affinity for the catalyst than the alkene products, resulting in exceptional Z-selective performance. The reviewer has thoroughly checked the manuscript. The work is very interesting. A good substrate scope and generality of the catalytic protocol are reported. The manuscript is nicely written. However, there is serious drawbacks are following:

Comments: *1. In the introduction part the authors nicely mentioned the three pincer Co and Fe complexes have been developed to facilitate this transformation with favorable Z-selectivity (Fig. 1a). These catalysts are well defined. However, in the current work, the catalyst is not well defined. Have the authors isolated the actual cobalt complex out of the reaction between Co(acac)₂ and a CNC-iPr ligand? What is the structure of this complex? During the reaction, Cobalt gets oxidized to from +2 to +3 state. A detailed understanding of the structure and bonding is required to define the catalyst.*

Response: We thank the reviewer for the comments. Over the past two years, we have been attempting to obtain more detailed information about the catalyst. However, due to the high activity of the catalyst, we have been unable to isolate it. We have tried hundreds of methods, involving different gas atmospheres, solvent combinations, temperatures, and auxiliary ligands, but have been unable to directly obtain crystals of Co(II) and the ligand. When using Et₂O/MeOH, EA/DMF, or EA/DMSO as solvents, we obtained the crystal of Co(III) and the ligand (Fig. 5e). This crystal can be directly used in catalytic reaction and exhibit a similar reactivity, representing a resting state of the actual catalyst. Their structure directly demonstrates the coordination relationship between the metal Co and the ligand. Combined with the HRMS results (Fig. 5d), it can be inferred that the binding mode of Co(II) and the ligand is as depicted in Fig. 5d, with anions serving as X-type ligands directly coordinating with Co, which is also a common form of complexes between tridentate ligands and Co(II) (please see references 56-61, and: Zhang, G., Vasudevan, K. V., Scott, B. L., Hanson, S. K., Understanding the Mechanisms of Cobalt-Catalyzed Hydrogenation and Dehydrogenation Reactions. *J. Am. Chem. Soc.* **135**, 8668–8681 (2013).; Mastalir, M., Tomsu, G., Pittenauer, E., Allmaier, G., Kirchner, K. Co(II) PCP Pincer Complexes as Catalysts for the Alkylation of Aromatic Amines with Primary Alcohols *Org. Lett.* **18**, 3462–3465(2016).; Junge, K., Papa, V., Beller, M. Cobalt–Pincer Complexes in Catalysis. *Chem. Eur. J.* **25**, 122–143 (2019).).

Mechanistically (Fig. 5f), Co(II) is not oxidized to Co(III) directly, but under reductive conditions, Co(II) is first reduced to a Co(I)-H complex **I** (this complex has been confirmed by HRMS (Fig. S17), and this process is also a common occurrence for Co(II) in a reductive condition, as seen in the references 38, 56-61, 63, and then undergoes ligand exchange with the alkyne to form **II**,

followed by oxidative addition with HBpin to obtain the Co(III) complex **III**. To avoid misunderstanding, we have indicated the oxidation states of Co in the revised Fig. 5f.

Comments: 2. *While this work shows the use of a new cobalt catalytic system, but several other similar methods exist to generate a similar type of Z-selective hydroboration, but it is not clear what the advantage of this method is over previously published examples. Cobalt catalyst is already used for Z-selective hydroboration (Reference 37).*

Response: Compared to the reported catalytic systems, our catalytic system has several significant advantages, as noted by the first reviewer: "the method is a significant improvement over the state of the art as reported." Firstly, catalytic efficiency is a core indicator of the superiority of a catalyst. In terms of catalytic efficiency, our catalytic system's catalytic activity far surpasses all previously reported alkyne hydroboration catalysts, including precious metal catalysts. In fact, the turnover frequency (TOF > 132,000 h⁻¹) exhibited by this catalytic system represents a limit for transition metal-catalyzed reactions, and we have not found any homogeneous catalytic reaction for comparison. Secondly, in terms of the substrate scope for Z-selective hydroboration reactions, our catalytic system far exceeds the substrate scope of other catalytic systems. Other catalytic systems are either limited to aryl alkynes, vinyl alkynes, or alkynes with unique electronic properties. Furthermore, since these catalytic systems commonly use bulky ligands, they are not suitable for bulky substrates. We are the only ones exploring Z-selective hydroboration of bulky alkyne substrates. Additionally, the Z-selectivity of the reported catalytic systems varies, and our system achieves excellent selectivity for all substrates. Thirdly, our catalytic system can achieve the complete conversion of substrates to Z-alkenes, followed by complete conversion to E-alkenes. This unique catalytic process, which changes configuration solely through extended reaction times, provides an excellent template reaction for kinetic studies. In these respects, our catalytic system is far superior to the Co-catalyzed system reported in reference 38, which only suitable for a very limited scope of substrates with low catalytic activity (TON up to 30, TOF up to 5 h⁻¹). Moreover, all reported catalytic systems require multi-step synthesis of ligands, which are difficult to obtain, significantly limiting their practical use. Our ligands can be obtained in a single step from inexpensive commercial materials with excellent yields. We are also the only ones demonstrating the convenient large-scale synthesis (30 mmol) of a catalytic system, making it highly applicable for synthesis.

Comments: 3. *Authors reported that the longer reaction time was directed to the E isomer, but the exact reaction time and mechanism were not optimized. But it would be interesting to know how isomerization occurs and what is the role of catalyst.*

Response: The process of conversion has been clearly demonstrated in the kinetic study section: the substrate is first completely converted to Z-alkene products, and then the Z-alkene products are further converted to E-alkene products, as detailed in Fig. 4b. The fundamental reason for this unique phenomenon is that the catalyst can better coordinate with the alkyne substrate compared to the alkene products. Therefore, until the alkyne substrate is completely reacted, the catalyst does not coordinate with the Z-alkene substrate to undergo Z to E configurational isomerization. Evidence supporting this point can be found in Fig. 5b. Under the provided conditions, compound

3a would normally be fully converted to *Z*-alkene in 5 seconds and then begin to convert to *E*-alkene (please see Fig. 4d). However, due to the presence of inert alkynes occupying the coordination sites of the catalyst, the *Z/E* isomerization did not significantly occur. During the *Z/E* isomerization process, the Co-H complex first inserts into the *Z*-alkene and then undergoes beta-H elimination to yield the thermodynamically more stable and less sterically hindered *E*-alkene. This is a typical process of *Z-E* isomerization for a Co catalyst (for examples, see references 56, 60, 61, 64, 66). This process has also been confirmed through our deuterium labeling experiments clearly (Fig. 5a, eq 1; Fig. S5).

Fig. 4b Kinetic profile of the hydroboration of phenylacetylene

Fig. 5b Inhibition of *Z/E* isomerization with an inert alkyne.

Fig. 5a, eq 1

Fig. S5

Comments: 4. They used base *t*BuOK 5.6 equiv. with respect to [Co] catalyst but what is the role of excess base is not clear in the mechanism.

Response: The CNC ligand contains two carbene sites, necessitating the use of two equivalents of *t*BuOK for hydrogen removal. Considering the ligand's equivalence to 1.4 equivalents of the catalyst, a minimum of 2.8 equivalents of *t*BuOK is required. To ensure optimal activation of the carbene in situ in the solvent, an excess of base was utilized to mitigate experimental errors, particularly when employing a low catalyst loading.

Comments: 5. Authors write “few catalytic reactions leading selectively to the *Z*-isomers have been reported”. This statement is not correct. There are many reports.

Response: The phrase “a few catalytic reactions” is presented in our manuscript. It is possible that the letter “a” was inadvertently omitted during the revision or uploading process. The description refers to the catalytic hydroboration reactions of terminal alkynes to form the *Z*-isomers. The total number of the references is 12 (references 32-43). To avoid misunderstanding, the sentence has been revised as “only a dozen of catalytic reactions leading selectively to the *Z*-isomers through hydroboration of terminal alkynes have been reported”.

Comments: 6. Authors should cite the following article.

Yuqing Zhang, Yanhui Chen, Qingyu Tian, Binju Wang, Guolin Cheng. Palladium-Catalyzed Multicomponent Assembly of (*Z*)-Alkenylborons via Carbopalladation/Boronation/Retro-Diels–Alder Cascade Reaction. *The Journal of Organic Chemistry* 2023, 88 (16), 11793-11800. <https://doi.org/10.1021/acs.joc.3c01084>

Response: This article has been cited as reference 20

Comments: 7. ¹¹B NMR of the products should be provided for all the organoboron compounds.

Response: All data has been submitted in accordance with the journal's guidelines. Comprehensive $^1\text{H-NMR}$, $^{13}\text{C-NMR}$, $^{11}\text{B-NMR}$, $^{19}\text{F-NMR}$, and HRMS data are provided for all of the new compounds. Additionally, $^1\text{H-NMR}$ data and spectra for known compounds were included and were found to be consistent with the reported information, with appropriate references cited. The $^{11}\text{B-NMR}$ data and spectra of the known compounds **1b**, **3b**, **8b**, **10b**, **11b**, **12b**, **13b**, **28b**, **31b**, **40b**, **41b**, **42b** have been added in the revised manuscript.

Comments: 8. Have the authors used Catechol boron (BCat) or other boranes as the boron source?

Response: As suggested, we have tested other boron compounds such as HBCat, HBdan, and 9-BBN. However, none of them showed reactivity. The results have been added to the revised SI as Table S12

Table S12 Screening of other boranes

Entry	H-[B]	Yield
1	HBCat	0
2	HBdan	0
3	9-BBN	0

HBCat

HBdan

9-BBN

Comments: 9. In SI Fig. S2, is not clean. A better picture should be given.

Response: As suggested, the Fig. S2 has been updated.

Fig. S2. The original kinetic data of **1a** with different cobalt concentrations.

Comments: 10. *The mechanism of the reaction is not very convincing.*

Response: Based on mechanistic experiments and previous related research, the mechanism is actually very clear. We'd like to explain it through a combination of experimental evidence and the proposed mechanism. The process from the catalyst precursor to intermediate **I** ($Co-H$) has been detailed previously. This process is quite common under reductive reaction conditions (see references 38, 56-61, 63), and the existence of intermediate **I** has been confirmed through HRMS analysis (Fig. S17). As the hydrogen on $Co-H$ is considered hydridic, and the hydrogen on the alkyne is considered acidic, intermediate **I** reacts with the terminal alkyne to produce hydrogen gas and intermediate **II**. This process is also a common catalytic process, and the generated hydrogen gas has been detected in the gas phase (Fig. S18), providing validation. Intermediate **II** undergoes oxidative addition with $HBpin$ to form intermediate **III**, which then undergoes reductive elimination to yield intermediate **IV**. After a syn-insertion of $Co-H$, intermediate **V** is obtained, followed by a

reaction with the acidic alkyne hydrogen to yield Z-alkene products and regenerate intermediate **II**. Each step in this cycle is a conventional and classical process, without any special reaction steps. Particularly noteworthy is the source of the product alkene hydrogen, as evidenced by the reaction with deuterated reagents (Fig. 5a), with a clear trajectory (hydrogens labeled in red and blue), providing strong support for the primary processes of the catalytic cycle. The Z/E isomerization begins only after the consumption of the alkyne substrate, primarily because the alkyne binds more effectively to the Co catalyst. This conclusion is directly observed through kinetic experiments (Fig. 4b, Fig. 5b). The steps of Z/E isomerization occur through Co-H insertion followed by syn-elimination, which is also a classic process, and has been clearly confirmed through deuterium labeling experiments (Fig. 5a, eq1, and Fig. S5). We hope that our explanation and revisions will satisfy the reviewer.

Comments: *Overall in the current manuscript, the novelty of the work is missing and a major revision is recommended for publication in Nature Communications.*

Response: The novelty of our work and the superiority of our system over other reported catalytic systems have been elucidated in our previous responses. The manuscript has been comprehensively revised according to the comments of the reviewer. We hope that our response and revisions will meet the satisfaction of the reviewer. We have expressed our gratitude to the reviewer in the acknowledgments section.

REVIEWER COMMENTS

Reviewer #1 (Remarks to the Author):

The revised manuscript by Peng and co-workers (manuscript ID: #NCOMMS-23-44051A), contains additional experiments and clarifications regarding their earlier submission (#NCOMMS-23-44051), where they described the efficient and highly stereoselective hydroboration of terminal alkynes.

After carefully reviewing the manuscript, I am of the opinion that this current study indeed has become suitable for publication in nature communications. The comments and concerns raised by this reviewer has been appropriately addressed. The reviewer regrets that computational studies were not included, but understands the reasoning behind it.

Overall, I recommend this manuscript for publication in nature communications.

However, one final comment remains that should be addressed in the manuscript:

The role of HBPin in the mechanism of the reaction is not entirely clear. Obviously from the experiments increasing the equiv. of HBpin accelerates the reaction and/or the Isomerization from Z to E. However, Z to E isomerization is independent on HBpin, while it's unclear, mechanistically, how HBpin accelerates the overall hydroboration reaction (Table S5). This warrants and explanation either through further investigation/experiments or a statement that the exact role of HBpin remains unclear.

Reviewer #2 (Remarks to the Author):

[Editor's note: In comments to the editorial office, the reviewer remarked that their concerns were addressed, and the manuscript is now suitable for publication.]

Point-by-point responses to the comments of reviewers

Reviewer #1 (Remarks to the Author):

Th The revised manuscript by Peng and co-workers (manuscript ID: #NCOMMS-23-44051A), contains additional experiments and clarifications regarding their earlier submission (#NCOMMS-23-44051), where they described the efficient and highly stereoselective hydroboration of terminal alkynes.

After carefully reviewing the manuscript, I am of the opinion that this current study indeed has become suitable for publication in nature communications. The comments and concerns raised by this reviewer has been appropriately addressed. The reviewer regrets that computational studies were not included, but understands the reasoning behind it.

Overall, I recommend this manuscript for publication in nature communications.

However, one final comment remains that should be addressed in the manuscript:

Comments: *The role of HBPin in the mechanism of the reaction is not entirely clear. Obviously from the experiments increasing the equiv. of HBpin accelerates the reaction and/or the Isomerization from Z to E. However, Z to E isomerization is independent on HBpin, while it's unclear, mechanistically, how HBpin accelerates the overall hydroboration reaction (Table S5). This warrants and explanation either through further investigation/experiments or a statement that the exact role of HBpin remains unclear.*

Response: We appreciate the positive comments of the reviewers. As mentioned by the reviewers, HBpin does not participate in the Z/E isomerization of the olefin. The increased presence of HBpin accelerates the reaction for two reasons. Firstly, it leads to a higher concentration of the boron reagent, thereby kinetically accelerating the formation of the Z-isomer. Since the reaction initially generates the Z-isomer before the E-isomer, the overall effect is an acceleration of the reaction with increased HBpin. The comprehensive demonstration of this phenomenon is evident from the contrasting reaction performance in Table S4 and the main text (Fig. 2 and Fig. 4d) under varying amounts of HBpin. The use of CoCl₂ as a suboptimal catalyst precursor in Table S4 highlights the pronounced effect of HBpin on reaction enhancement. Additionally, the Z/E isomerization is catalyzed by the Co-H complex, which is unstable and prone to deactivation. HBpin serves as a prerequisite for generating the Co-H species and providing the necessary reductive conditions for its stabilization (Fig. 4a, eq 1, right). Therefore, an increased amount of HBpin facilitates the generation and stabilization of a sufficient concentration of the Co-H species, thereby accelerating the Z/E isomerization. In order to clearly articulate the role of HBpin, we have included the following statements in the main text: "Therefore, the increased presence of HBpin can significantly promote the reaction process due to kinetic effects (Table S4, see also Fig. 2)," and "An appropriate excess of HBpin can ensure the generation of a sufficient concentration of the Co-H complex and provide a reductive environment to stabilize it." These sentences have been highlighted in "manuscript-revision highlighted in yellow".

REVIEWERS' COMMENTS

Reviewer #1 (Remarks to the Author):

The answers of the authors and subsequent modification of the manuscript have positively addressed my remaining concern.

I recommend publication in Nature Communications